# Association of Dietary Inflammatory Potential with Blood Inflammation: The Prospective Markers on Mild Cognitive Impairment

**DOI:** 10.3390/nu14122417

**Published:** 2022-06-10

**Authors:** Xuan Wang, Tiantian Li, Hongrui Li, Dajun Li, Xianyun Wang, Ai Zhao, Wannian Liang, Rong Xiao, Yuandi Xi

**Affiliations:** 1Beijing Key Laboratory of Environmental Toxicology, School of Public Health, Capital Medical University, Beijing 100069, China; 122020010126@ccmu.edu.cn (X.W.); ltt0512@mail.ccmu.edu.cn (T.L.); lhr2020144@ccmu.edu.cn (H.L.); lidajun@ccmu.edu.cn (D.L.); wangxy0504@mail.ccmu.edu.cn (X.W.); xiaor22@ccmu.edu.cn (R.X.); 2Wanke School of Public Health, Tsinghua University, Beijing 100084, China; aizhao18@tsinghua.edu.cn (A.Z.); liangwn@tsinghua.edu.cn (W.L.)

**Keywords:** inflammation, mild cognitive impairment, energy-adjusted dietary inflammatory index, systemic immune inflammation index, system inflammation response index

## Abstract

Inflammation is known as an important mechanism of cognitive dysfunction. Systemic immune inflammation index (SII) and system inflammation response index (SIRI) are two blood inflammatory markers, which are related to many chronic diseases including cognitive impairment. It is recognized that dietary inflammatory index (DII), which is used to estimate the overall inflammatory potential of diet, may be related to mild cognitive impairment (MCI) as well. This study aimed to explore the relationship between SII, SIRI and DII, as well as the role of these inflammatory indexes on MCI in elderly people. A total of 1050 participants from Beijing were included. Neuropsychological tests were used for cognitive evaluation. Energy-adjusted DII scores were calculated based on semi-quantitative food frequency questionnaire. Blood samples were tested for calculating SII and SIRI. Log-binomial regression models were used to estimate the correlation of indexes. After adjusting demographic characteristics, SII and SIRI in MCI individuals were higher than controls (*p* ≤ 0.001). DII, SII and SIRI had positive relationship with MoCA scores (*p* < 0.005). DII also correlated with SIRI in MCI (β = 0.11, *p* = 0.031). Higher DII and SIRI could definitely increase the risk of MCI, as well as DII and SII (*p* < 0.005). In conclusion, DII was positively correlated with blood inflammation. The elderly with higher level of DII and SIRI, or DII and SII could be considered as people with higher risk of developing MCI.

## 1. Introduction

The world’s population is aging rapidly. Substantial increases in the number and proportion of older people have been recorded in virtually every country in the world [1]. It is undeniable that the social burden of age-related cognitive decline and dementia is expected to increase [2]. The functional shift in the immune system towards a proinflammatory phenotype is a common characteristic of aging [3,4]. Acute inflammation could help heal wounds and promote tissue regeneration. When this necessary process of the body’s natural response to tissue injury is not controlled properly, a chronic, low-grade inflammatory state will appear [5]. Scientific evidence has suggested that this kind of inflammation plays a critical role in the pathogenesis of cognitive decline and dementia [4,6].

As a modifiable lifestyle factor, the consumption of various foods and nutrients, such as MUFA, PUFA, amino acids, thiamine, anthocyanins and vitamins, may be a valuable contributor to regulating systemic inflammation of human body [7,8,9]. As daily diets rather than single nutrients play comprehensive effects [10], it is important to explore the relationship between comprehensive diet index and neurological disorders. The dietary inflammatory index (DII) [5] was built in nutritional research to characterize and measure the overall inflammatory potential of the diet. It could help develop precise cognitive health maintenance strategies or tailor useful dietary interventions. There is evidence that high DII scores and low anti-inflammatory dietary scores are correlated with low global cognitive function and high risk of mild cognitive impairment (MCI) [11,12,13,14]. However, other studies have shown that the DII score is not correlated with human brain structure or cognitive function [15]. It is necessary to clarify the relationship between diet-related inflammation and cognitive health.

Blood inflammatory indexes are inexpensive and accessible biomarkers. Evidence has shown that platelet, platelet–lymphocyte ratio (PLR) or neutrophil–lymphocyte ratio (NLR) are associated with risk of stroke and cardiovascular diseases [16,17]. Two novel inflammatory markers, system inflammation response index (SIRI) and systemic immune inflammation index (SII), evaluated by platelet and three subtypes of leukocyte, have been proposed to be associated with atherogenesis [18] and cardiovascular diseases [19]. However, the impact of SIRI and SII on cognitive function still needs to be explored. The synergistic effect of SII, SIRI and DII on cognitive health should be detected as well. The aim of the present study was to explore the correlation of DII on SIRI and SII, while demonstrating the prospective role of these inflammatory markers on the risk of developing MCI.

## 2. Materials and Methods

### 2.1. Participants

Participants aged from 65 to 85 years old were collected from 2020 to 2021 in different centers of Beijing (registered at Chinese Clinical Trial Registry as ChiCTR2100054969). The workflow and standards were adopted as previous study [20,21]. A total of 1050 participants were collected. This study was conducted in accordance with the principles of the Declaration of Helsinki and ethically approved by the Ethics Committee of Capital Medical University (Z2019SY052). All subjects have signed informed consent before they were included.

### 2.2. Cognitive Assessment

Face to face tests were conducted in a standard order by trained personnel in a quiet room. Cognitive function was assessed by the Montreal Cognitive Assessment (MoCA). The cutoff points of MoCA were determined according to education level. Mini-mental state examination (MMSE) score was also used to exclude any subject with AD [22]. Two-stage procedure was used to diagnose MCI patients according to our previous study [21]. Briefly, participants were suspected of having MCI based on MoCA performance, and they had a secondary examination by neurologists to confirm the clinical diagnosis.

### 2.3. Dietary Assessment

The information of dietary intake was collected by food frequency questionnaire (FFQ) of 2002 China National Nutrition and Health Survey (CNHS 2002) [23], which obtained the habitual intake of foods over the past year. The intake of energy and nutrients were calculated by using the China Food Composition Database (Version 6) [24].

### 2.4. Calculation of DII

The dietary inflammatory index was used to assess inflammatory potential of diet according to Shivappa et al. [25]. The 23 out of the 45 food parameters in the original DII were used in this study, including 7 proinflammatory parameters (energy, carbohydrate, protein, total fat, saturated fatty acids, cholesterol, and Ferrum (Fe)) and 16 anti-inflammatory parameters (monounsaturated fatty acid (MUFA), polyunsaturated fatty acids; fiber; Magnesium (Mg); niacin; thiamine; riboflavin; vitamins A, C and E; beta carotene; zinc (Zn); selenium; anthocyanin; isoflavones and grams of alcohol) [25]. Participants who consumed alcohol over 40 g/d were excluded, because more than 40 g/d consumption of alcohol cannot exert any anti-inflammatory effects [26]. To reduce the bias induced by different energy intake, the energy-adjusted DII was calculated in accordance with the published procedures [27,28]. First, energy-adjustment was performed using the density method [29]; all of the food/nutrients and the global database were converted to units per 1000 kcal [21,30]. Next, a z-score was created by subtracting the global standard mean from the individual’s estimated intake and dividing by its standard deviation [27]. Furthermore, this value was converted into a centered proportion score, with values ranging from 0 to 1, in order to minimize the effect of right skewing [31]. The proportion was then centered by doubling the proportion and subtracting 1 [12]. Finally, the personal DII score was obtained by adding up all dietary component scores. A high (i.e., more positive) score indicates the diets may have higher pro-inflammatory properties [28].

### 2.5. Laboratory Measurements

Blood samples were collected from fasting participants in the morning. The absolute peripheral counts of neutrophils, lymphocytes, monocytes and platelets were analyzed by automated analyzer. The systemic immune–inflammation index was calculated from absolute peripheral platelet counts (P; ×10^9^/L), neutrophil counts (N; ×10^9^/L), monocyte counts (M; ×10^9^/L) and lymphocyte counts (L; ×10^9^/L) by applying the formula: SII = P × N/L, SIRI = N × M/L [32]. We removed participants who had cell counts beyond the clinical reference ranges (i.e., those who had a neutrophil count <2.0 or >7.5 × 10^9^/L (N = 63); those who had a lymphocyte count <0.8 or > 5.0 × 10^9^/L (N = 9); and those who had a platelet count of less than <100 or >300 × 10^9^/L (N = 101)) to test the robustness of the findings [18].

### 2.6. Statistical Analysis

Continuous variables were expressed as medians (interquartile ranges, IQR) when they were non-normally distributed; otherwise, a mean ± standard deviation (SD) was used in the normal distribution. Analysis of variance (ANOVA) or the Kruskal–Wallis rank test was applied for continuous variable, while Chi-squared was used for categorical variable. SII, SIRI and DII scores were initially treated as continuous variables. In the log-binomial regression, SIRI and SII were divided into four groups. DII was categorized into three groups according to other study [33]. Log-binomial regression models were adjusted by gender, age, education levels, smoking, BMI groups and BMR groups. Statistical significance was set at a two-sided *p* < 0.05. All statistical analyses were performed by using IBM SPSS Statistics 26 (IBM Corp., Armonk, New York, USA).

## 3. Results

### 3.1. Demographic, Clinical Characteristics and Dietary Intake of Participants

The characteristics were described in Table 1. Of 1050 participants, 59.9% were female. There were no differences of age, body mass index (BMI) or basal metabolic rate (BMR) between MCI and control individuals. However, the percentages of males, smokers and higher educated people in the MCI group were higher than in the control group. In Figure 1, some anti-inflammatory nutrients and blood inflammatory markers were detected in different groups. The consumption of MUFA, Fe, Mg, Zn, vitamin E, thiamine, anthocyanins and isoflavones were lower in MCI individuals than in controls. Lymphocyte, as a subtype of leukocyte, was at a lower level in the MCI group (Figure 2).

### 3.2. Comparisons of DII, SII, SIRI between MCI and Controls

Subsequently, DII, SIRI and SII were calculated. IQR score for DII was 0.8 (−0.1, 1.4) with a range from −3.796 to 3.904. SIRI was 0.64 (0.46, 0.90), ranging from 0.158 to 3.005. SII was 396.63 (295.28, 527.68), ranging from 94.244 to 1732.707. Although the difference of DII was not found, MCI patients had definitely higher SIRI (*p* < 0.001) and SII (*p* = 0.001) than controls as expected (Table 2).

### 3.3. Correlation of DII, SII, SIRI with MoCA Score

Multiple linear regression was used to explore the correlation of DII, SIRI, SII with cognitive score (Table 3). DII (β = −0.363, *p* = 0.007) and SIRI (β = −1.505, *p* < 0.001) were shown negatively associated with MoCA score after adjustment for age, gender, education, smoking, BMI and BMR. Meanwhile, DII (β = −0.394, *p* = 0.005) and SII (β = −0.003, *p* = 0.001) were found negatively correlated with MoCA score after adjustment for the same characteristics (Table 4).

### 3.4. Performance of DII on SIRI and SII in MCI Patients

The relationship between DII and blood inflammation index (SIRI or SII) in MCI patients was further explored by multiple linear regression (Table 5). DII was found to be positively correlated with SIRI (β = 0.042, *p* = 0.031) after adjustment for gender, age, education, smoking, BMI and BMR. There was no correlation between DII and SII (β = 10.811, *p* = 0.298).

### 3.5. The Roles of Inflammatory Markers on Suffering from MCI

Generalized linear regression was conducted to explore the association between inflammatory scores and the risk of MCI. Q1 was the reference in all comparisons. First, DII, SIRI and SII were tested separately after adjustment for age, gender, education, smoking, BMI and BMR (Table 6). Results showed Q3 of DII was the risk factor of MCI (prevalence ratio (PR) = 1.23, 95% confidence interval (CI): 1.03, 1.47). Q3 and Q4 of SIRI could increase the risk of MCI (PR = 1.28, 95% CI: 1.04, 1.57; PR = 1.31, 95% CI: 1.07, 1.60). Q4 of SII was also correlated with higher risk of MCI (PR = 1.29, 95% CI:1.06, 1.57). In addition, both DII and SIRI were tested in Model 1, as well as DII and SII in Model 2. Higher DII and SIRI could definitely increase the risk of MCI (*p* < 0.005). The same results were found in DII and SII (*p* < 0.005) as well (Table 7).

## 4. Discussion

In elderly people, immune system features begin to reveal decline because of senescence [34]. In this case, chronic inflammation begins to adversely affect human aging, possibly contributing to the development and clinical course of age-related conditions such as cardiovascular, metabolic and neurodegenerative diseases [35]. There is evidence that diet may affect the inflammatory condition of the human body. DII is recognized as a dietary inflammatory indicator which combines the anti-inflammatory/pro-inflammatory effects of multiple food ingredients. The relationship between the DII and metabolic syndrome, which is the common risk factor for chronic disease, has been verified [36]. SIRI and SII are recognized more reliable and more representative to reflect inflammation and thrombosis than PLR and NLR [37,38]. However, there is little evidence of the correlation between DII and SII or SIRI and of its effects on MCI. Our results manifested that DII, SIRI and SII had a positive relationship with MoCA scores. DII was positively correlated with blood inflammation. The elderly with a higher level of DII and blood systemic immune–inflammation index (SIRI or SII) had a higher risk of developing MCI.

Results of the present study showed that males, smokers and higher educated people had higher risk of suffering from MCI. This is similar to other studies [39,40]. There is evidence indicating that higher education is associated with memory, executive function and language dysfunction in dementia patients and is related with faster cognitive decline on global cognition [41]. MUFA, Mg, Fe, Zn, vitamin E, thiamine, anthocyanins and isoflavones, which are recognized as anti-inflammatory parameters [42,43,44,45,46], were found to be much less consumed by MCI patients. Lymphocytes, as a key inflammatory response regulator, were down-regulated in MCI individuals. Evidence indicates lower lymphocytes are highly associated with stroke [47]. The higher level of lymphocytes within a normal range plays a vital role in immune tolerance and homeostasis [48].

The SII is believed to be an indicator of stroke and cardiovascular diseases [18], and it is similar to ours. Patients with acute ischemic stroke (AIS), which is usually accompanied by severe symptoms of neurological deficit, tend to have a higher SII compared with healthy controls [38,49]. There is evidence indicating that elevated SIRI and SII could increase the risk of stroke, stroke subtypes and all-cause death [19]. Recently, SIRI was also determined to be an independent predictive indicator for intracerebral hemorrhage [50] and AIS [51] functional outcomes. The results of the present study suggested that MCI people had higher scores of SIRI and SII; DII was correlated with SIRI in MCI patients; DII, SIRI and SII were negatively correlated with the MoCA score. A recent study explored the upregulation of inflammatory biomarkers such as TNF-α and IL-1β in people with cognitive decline [52]. Our results might provide a new perspective on this relationship.

In view of the relationship between DII and SIRI in the present study, DII might be contribute to the development of chronic inflammatory processing in aging. DII has been reported to have association with numerous plasma inflammatory markers (CRP and IL6) and multiple health outcomes (stoke, depression, CVDs, metabolic risk markers, cancer and all-cause and specific-cause mortality) [53]. Recent studies found that DII is negatively associated with the Mediterranean dietary pattern and is positively correlated with the PLR, NLR and SII among polycystic ovary syndrome patients [31]. Our study validated the positive role of DII on SIRI. It provided new evidence to explain the association between dietary inflammation and blood inflammation.

Some researchers have reported the correlation between DII and cognitive function. The cohort study indicates that a proinflammatory diet is significantly associated with cognitive decline [54]. The association of DII and neurological disorders including memory function and dementia has been proved [11]. Cross-sectional studies demonstrate that the DII score is negatively correlated with global cognitive function, verbal memory [12], episodic memory, working memory and semantic memory [11] in different countries. In a randomized, double-blind, placebo-controlled trial, higher DII scores were correlated with cognitive decline in women aged 65–79 [33]. Furthermore, the anti-inflammatory values of the DII were also correlated with the length of the telomere [25]. The results of the present study demonstrated that DII, SIRI and SII were negatively correlated with the MoCA score separately. Moreover, when DII combined different systemic immune–inflammation indexes (SII and SIRI), the results of association in inflammation and the MoCA score were similar. It manifested that DII, SIRI and SII could be synergistically estimated as the critical factors influencing cognitive function.

Generalized linear regression was used to further explore the synergistic effect of SII, SIRI and DII on the risk of MCI. Results showed that higher DII and SIRI could increase the risk of MCI definitely. Meanwhile, DII and SII showed similar results. Compared to analyzing these inflammatory markers respectively, comprehensive evaluation of dietary and blood inflammation could contribute to finding people at higher risk of cognitive decline, developping more precise cognitive health maintenance strategies and customizing effective dietary intervention programs as well.

This study also had a few limitations. First, components for calculating DII were incomplete. Second, recent studies provide interesting points that the effect of proinflammatory parameters of DII in cognitive decline might be associated with deterioration of nutrition [7,9]. It is worthy to explore more associations among the irrational nutrition, blood inflammation markers and MCI. Last but not least, it is difficult to draw causal inferences regarding the etiological links between DII, SII, SIRI and MCI in a cross-sectional design. More studies are warranted to replicate our work and verify the conclusions.

## 5. Conclusions

Our study found the potential effect of DII, SIRI and SII on MCI. DII, SIRI and SII had positive relationship with the MoCA score. Furthermore, new proof was provided of a positive relationship between DII and blood inflammation. Finally, the elderly with a higher level of DII and SIRI, or DII and SII could be considered as people with a higher risk of developing MCI. These results supplemented vital evidence that dietary and blood inflammatory markers have an important synergistic impact on MCI.

## Figures and Tables

**Figure 1 nutrients-14-02417-f001:**
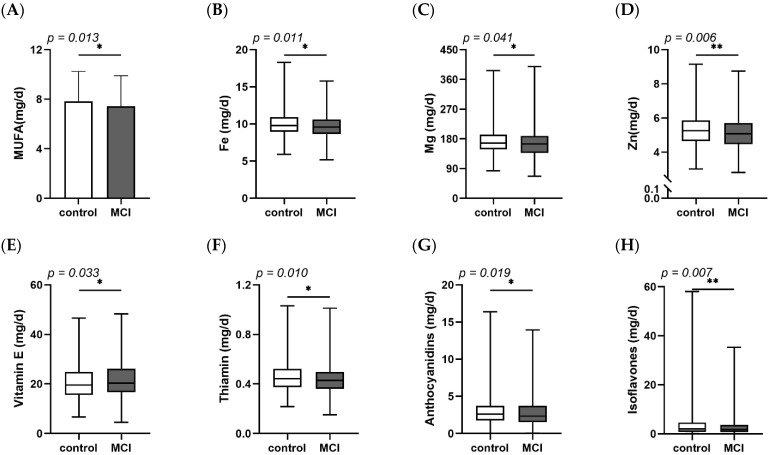
Dietary intake of (**A**) MUFA, (**B**) Fe, (**C**) Mg, (**D**) Zn, (**E**) Vitamin E, (**F**) Thiamin, (**G**) Anthocyanidins and (**H**) Isoflavones between control and MCI. Fe, Ferrum; MCI, mild cognitive decline; MUFA, monounsaturated fatty acids; Mg, Magnesium; Zn, Zinc. * *p* < 0.05, ** *p* < 0.01.

**Figure 2 nutrients-14-02417-f002:**
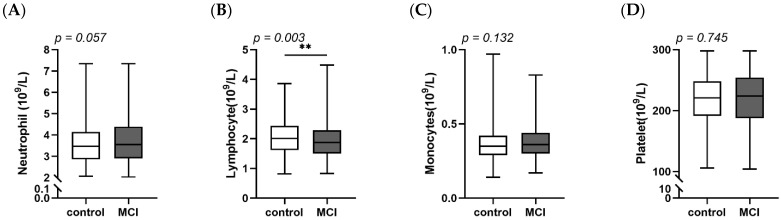
Concentration of inflammatory markers in blood between control and MCI. (**A**) Neutrophil counts; (**B**) Lymphocyte counts; (**C**) Monocytes counts; (**D**) Platelet counts. MCI, mild cognitive decline. ** *p* < 0.01.

**Table 1 nutrients-14-02417-t001:** Demographic characteristics overall and by MCI.

	Total	Group		*p*
		Control	MCI	
Demographic characteristics				
*N*	1050	481	569	
Age	70 (67.73)	70 (67.73)	69 (67.73)	0.759
Female *n* (%)	629 (59.9)	313 (65.1)	316 (55.5)	0.002 **
Smoking	210 (28.0)	75 (21.9)	135 (33.2)	0.001 **
MoCA score	21 (17, 23)	22 (20, 25)	19 (15, 22)	<0.001 **
BMR (kcal)	1258 (1154, 1387)	1263 (1157, 1377)	1254 (1149, 1395)	0.839
Education				<0.001 **
Illiterate *n* (%)	234 (22.3)	165 (34.3)	69 (12.1)	
Primary school *n* (%)	353 (33.6)	194 (40.3)	159 (27.9)	
Junior high school *n* (%)	376 (35.8)	86 (17.9)	290 (51.0)	
High school and above *n* (%)	87 (8.3)	36 (7.5)	51 (9.0)	
BMI				0.111
Emaciation *n* (%)	14 (1.3)	4 (0.8)	10 (1.8)	
Normal *n* (%)	278 (26.5)	120 (24.9)	158 (27.8)	
Overweight *n* (%)	458 (43.6)	204 (42.4)	254 (44.6)	
Obesity *n* (%)	300 (28.6)	153 (31.8)	147 (25.8)	

MCI, mild cognitive decline; BMI, body mass index; BMR, basal metabolic rate; MoCA, Montreal cognitive assessment score. ** *p* < 0.01.

**Table 2 nutrients-14-02417-t002:** Dietary and blood inflammatory index overall and by MCI.

	Total	Group		*p*
		Control	MCI	
DII	0.78 (−0.09, 1.45)	0.80 (−0.07, 1.38)	0.76 (−0.16, 1.52)	0.726
SIRI	0.64 (0.46, 0.90)	0.61 (0.44, 0.84)	0.68 (0.48, 0.94)	<0.001 **
SII	396.63 (295.28, 527.68)	367.89 (285.26, 497.78)	412.87 (311.10, 544.06)	0.001 **

MCI, mild cognitive decline; DII, dietary inflammatory index; SIRI, system inflammation response index; SII, systemic immune inflammation index. ** *p* < 0.01.

**Table 3 nutrients-14-02417-t003:** Association between DII, SIRI and cognitive function.

	β	95% CI	*p*
Age	−0.048	(−0.125, 0.028)	0.215
Gender	−0.346	(−1.484, 0.793)	0.551
Education	2.401	(2.033, 2.768)	<0.001 **
Smoking	−0.631	(−1.440, 0.177)	0.126
BMI (kg/m^2^)	−0.019	(−0.123, 0.085)	0.719
BMR (kcal)	0.004	(0.000, 0.007)	0.028 *
DII	−0.363	(−0.625, −0.101)	0.007 **
SIRI	−1.505	(−2.265, −0.745)	<0.001 **

CI, confidence interval; BMI, body mass index; BMR, basal metabolic rate; DII, dietary inflammatory index; SIRI, system inflammation response index. * *p* < 0.05, ** *p* < 0.01.

**Table 4 nutrients-14-02417-t004:** Association between DII, SII and cognitive function.

	β	95% CI	*p*
Age	−0.061	(−0.140, 0.019)	0.134
Gender	−0.278	(2.021, 2.798)	0.640
Education	2.409	(−1.662, −0.010)	<0.001 **
Smoking	−0.836	(−1.448, 0.891)	0.047 *
BMI (kg/m^2^)	−0.019	(0.000, 0.007)	0.726
BMR (kcal)	0.003	(−0.670, −0.117)	0.067
DII	−0.394	(−0.005, −0.001)	0.005 **
SII	−0.003	(−1.448, 0.891)	0.001 **

CI, confidence interval; BMI, body mass index; BMR, basal metabolic rate; DII, dietary inflammatory index; SII, systemic immune inflammation index. * *p* < 0.05, ** *p* < 0.01.

**Table 5 nutrients-14-02417-t005:** DII associated with SIRI or SII in MCI.

	SIRI			SII		
	β	95% CI	*p*	β	95% CI	*p*
MCI						
Age	0.008	(−0.003,0.019)	0.147	1.424	(−4.143, 6.991)	0.615
Gender	−0.202	(−0.372, −0.033)	0.019 *	14.399	(−74.559, 103.357)	0.750
Education	−0.075	(−0.135, −0.016)	0.013 *	−31.599	(−63.730, 0.531)	0.054
Smoking	0.116	(0.006, 0.227)	0.039 *	29.550	(−27.917, 87.016)	0.312
BMI (kg/m^2^)	0.010	(−0.005, 0.025)	0.200	−0.183	(−8.310, 7.943)	0.965
BMR (kcal)	0.000	(−0.001, 0.000)	0.541	0.032	(−0.233, 0.296)	0.814
DII	0.042	(0.004, 0.079)	0.031 *	10.811	(−9.611, 31.232)	0.298

MCI, mild cognitive decline; BMI, body mass index; BMR, basal metabolic rate; DII, dietary inflammatory index; SIRI, system inflammation response index; SII, systemic immune inflammation index. * *p* < 0.05.

**Table 6 nutrients-14-02417-t006:** The effects of DII, SII and SIRI on the occurrence of MCI by log-binomial regression.

	MCI	
	PR (95% CI)	*p*
DII effects		
Q2(0~2) vs. Q1(<0)	0.90 (0.77, 1.06)	0.196
Q3(>2) vs. Q1(<0)	1.23 (1.03, 1.47)	0.025 *
SIRI effects		
Q2(0.5~0.6) vs. Q1(<0.5)	1.02 (0.81, 1.28)	0.872
Q3(0.6~0.9) vs. Q1(<0.5)	1.28 (1.04, 1.57)	0.018 *
Q4(>0.9) vs. Q1(<0.5)	1.31 (1.07, 1.60)	0.010 *
SII effects		
Q2(295~396) vs. Q1(<295)	1.01 (0.81, 1.27)	0.913
Q3(396~528) vs. Q1(<295)	1.11 (0.89, 1.38)	0.340
Q4(>528) vs. Q1(<295)	1.29 (1.06, 1.57)	0.013 *

Data are all adjusted by age, gender, education, smoking, BMI and BMR. MCI, mild cognitive decline; DII, dietary inflammatory index; SIRI, system inflammation response index; SII, systemic immune inflammation index; MCI, mild cognitive decline; PR, prevalence ratio. * *p* < 0.05.

**Table 7 nutrients-14-02417-t007:** The synergistic effects of DII with SII or SIRI on the occurrence of MCI.

	MCI	
	PR (95% CI)	*p*
Model1		
DII effects		
Q2(0~2) vs. Q1(<0)	0.9 (0.77, 1.06)	0.217
Q3(>2) vs. Q1(<0)	1.22 (1.01, 1.48)	0.041 *
SIRI effects		
Q2(0.5~0.6) vs. Q1(<0.5)	1.03 (0.82, 1.29)	0.810
Q3(0.6~0.9) vs. Q1(<0.5)	1.28 (1.04, 1.57)	0.018 *
Q4(>0.9) vs. Q1(<0.5)	1.32 (1.08, 1.62)	0.006 **
Model2		
DII effects		
Q2(0~2) vs. Q1(<0)	0.91 (0.77, 1.08)	0.266
Q3(>2) vs. Q1(<0)	1.25 (1.01, 1.54)	0.041 *
SII effects		
Q2(295~396) vs. Q1(<295)	1.03 (0.83, 1.29)	0.767
Q3(396~528) vs. Q1(<295)	1.1 (0.89, 1.37)	0.381
Q4(>528) vs. Q1(<295)	1.32 (1.08, 1.62)	0.007 **

Data are all adjusted by age, gender, education, smoking, BMI and BMR. MCI, mild cognitive decline; DII, dietary inflammatory index; SIRI, system inflammation response index; SII, systemic immune inflammation index; MCI, mild cognitive decline; PR, prevalence ratio. * *p* < 0.05, ** *p* < 0.01.

## Data Availability

Data are available upon reasonable request.

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
