# Peer review of "Association of Dietary Inflammatory Potential with Blood Inflammation: The Prospective Markers on Mild Cognitive Impairment"

_nutrients, 2022, doi:10.3390/nu14122417_

Round 1

Reviewer 1 Report

The authors present very interesting data concerning the relationship between diet, systemic infalammation status and cognitive impairment. The study design is simple but very elegant. I suggest to integrate introduction and discussion with these very recent papers which provide interesting points of reflection for the discussion (Aquilani R et al. Is the Brain Undernourished in Alzheimer’s Disease? Nutrients. 2022; 14(9):1872. https://doi.org/10.3390/nu14091872; Aquilani R et al. Mini Nutritional Assessment May Identify a Dual Pattern of Perturbed Plasma Amino Acids in Patients with Alzheimer's Disease: A Window to Metabolic and Physical Rehabilitation? Nutrients. 2020 Jun 21;12(6):1845. doi: 10.3390/nu12061845. PMID: 32575805; PMCID: PMC7353235.) Moreover, the authors should clarify the methods of elòection of patients: dementia was excluded only with MMSE score? Was any etiological biomarker used? Did the patients undergo a complete neuropsychological assessment? The methods described in lines 100-110 are not clear, please explain better. Why were subjects with cell count beyond the clinical reference range excluded? It excludes any haematologic pathology, but introduces a selection bias. A cell count beyond the reference range is not in fact a proof of disease, but could still be an expression of immune system modulation.

Reviewer 2 Report

In this paper Wang and colleagues investigate the relationship between systemic immune inflammation index (SII), system inflammation response index (SIRI) and diet inflammatory index (DII) in elderly people. They also evaluate the possible correlation between these indexes and mild cognitive impairment (MCI), and propose that energy adjusted-DII (E-DII), SIRI and SII represent risk factor for developing MCI.

Although the aim of paper is very interesting, the soundness of results is reduced as the investigation doesn’t provide any mechanistic insight into the proposed correlations.

Further, several concerns need to be addressed.

First of all a more comprehensive characterization of blood inflammatory status is required to support conclusions. Systemic inflammation was assessed in terms of platelet, neutrophil, monocyte and lymphocyte counts, but no information is reported about cytokine and adipokines levels. This analysis is crucial for a good assessment of blood inflammation, particularly when the investigation focuses on dietary inflammatory potential and on MCI.

The presentation of results is quite confusing and must be improved. In particular Table 1 is difficult to read, maybe the use of histograms for several data could be helpful.

The paper require an extensive grammatical and syntactic editing.

Round 2

Reviewer 2 Report

The manuscript has been improved and all concerns have been addressed.

This manuscript is a resubmission of an earlier submission. The following is a list of the peer review reports and author responses from that submission.